# Development of Photoluminescent and Photochromic Polyester Nanocomposite Reinforced with Electrospun Glass Nanofibers

**DOI:** 10.3390/polym15030761

**Published:** 2023-02-02

**Authors:** Mahmoud T. Abdu, Tawfik A. Khattab, Maiada S. Abdelrahman

**Affiliations:** 1Metallurgical Engineering Department, Faculty of Engineering, Cairo University, Giza 12613, Egypt; 2Mechanical Engineering Department, College of Engineering, University of Bisha, P.O. Box 421, Bisha 61922, Saudi Arabia; 3Dyeing, Printing and Auxiliaries Department, National Research Centre, Cairo 12622, Egypt

**Keywords:** polyester resin, electrospinning, glass nanofibers, lanthanide-doped aluminate nanoparticles, persistent photoluminescence and photochromism

## Abstract

A polyester resin was strengthened with electrospun glass nanofibers to create long-lasting photochromic and photoluminescent products, such as smart windows and concrete, as well as anti-counterfeiting patterns. A transparent glass@polyester (GLS@PET) sheet was created by physically immobilizing lanthanide-doped aluminate (LA) nanoparticles (NPs). The spectral analysis using the CIE Lab and luminescence revealed that the transparent GLS@PET samples turned green under ultraviolet light and greenish-yellow in the dark. The detected photochromism can be quickly reversed in the photoluminescent GLS@PET hybrids at low concentrations of LANPs. Conversely, the GLS@PET substrates with the highest phosphor concentrations exhibited sustained luminosity with slow reversibility. Transmission electron microscopic analysis (TEM) and scanning electron microscopy (SEM) were utilized to examine the morphological features of lanthanide-doped aluminate nanoparticles (LANPs) and glass nanofibers to display diameters of 7–15 nm and 90–140 nm, respectively. SEM, energy-dispersive X-ray spectroscopy (EDXA), and X-ray fluorescence (XRF) were used to analyze the luminous GLS@PET substrates for their morphology and elemental composition. The glass nanofibers were reinforced into the polyester resin as a roughening agent to improve its mechanical properties. Scratch resistance was found to be significantly increased in the created photoluminescent GLS@PET substrates when compared with the LANPs-free substrate. When excited at 368 nm, the observed photoluminescence spectra showed an emission peak at 518 nm. The results demonstrated improved hydrophobicity and UV blocking properties in the luminescent colorless GLS@PET hybrids.

## 1. Introduction

The creation of glow-in-the dark glasses often involves the use of rare earth oxides. Smart glass windows and other types of glass products that produce light have been used in many lighting systems. Cheapness, photostability, non-toxicity, and high-quality photoluminescence are just a few reasons why rare earth oxides are so well suited for usage in a smart window [1,2]. A glass window can be customized to provide a variety of benefits, including protection from sun rays, excess heat, and even visible light. After being exposed to a light source and then having that source turned off, certain materials exhibit a phenomenon called persistent photoluminescence, characterized by sustained emission in the visible, near-infrared, and ultraviolet spectrums [3,4,5]. Controlling the intensity and lifetime of photoluminescence is an active area of scientific inquiry. Therefore, advanced visual smart windows, electronic displays, and photonics have become feasible. Both hosting and doping materials affect on afterglow period and color [6]. The photoluminescence efficiency of lanthanide-activated nanomaterials is found to be very high throughout a wide range of applications. Therefore, many optical instruments, such as light-emitting diodes, use nano-scale materials that have been activated by rare earth elements. The physical properties of smart materials can be altered in response to an external stimulus such as chemical, electromagnetic, or photochemical agents [7,8]. The ability of smart materials to respond to dangerous stimuli such as extremely high temperatures and toxic chemical agents has led to their incorporation into protective devices. It is possible for a specific substance to keep emitting light for a considerable amount of time after the light supply has been shut off due to a phenomenon known as persistent emission. When illuminated with ultraviolet light, photochromic materials tolerate a colorimetric switching between two optical states. When the excitation source is switched off, the photochromic agent returns to its original state [9,10,11]. Potential applications of the photochromic effect in the technological realm include sensors, ophthalmic lenses, displays, and sunglasses. Security barcodes, trademark protection, ultraviolet shielding, military camouflage, and smart fabrics are just a few of the many uses that have been reported for our enhanced understanding of light-induced chromic materials [12,13,14].

The majority of commercialized photochromic agents are organic dyestuffs, such as *Spiropyrans* and *diarylethenes* [15,16]. However, there are certain downsides to organic photochromic colorants, including high costs and poor photostabilty. For the reason that the photochromism of such organic dyestuffs is based on the molecular switching of their structure, their inclusion in bulk materials limits their molecular switching ability due to steric hindrance effects [17]. This means that the photostability of photochromic organic dyes can be diminished with extended exposure to UV light. However, inorganic photochromic agents are resistant to steric effects as they do not depend on a change in their molecular structural system. Thus, inorganic photochromic agents provide better photochromism and higher photostability as compared with organic agents [18]. Eu^2+^ and other activation lanthanide ions are well-known photon emitters, particularly stimulated by UV light [19]. Inorganic compounds with photoluminescent lanthanide activators have been found to exhibit novel optical, electrical, and magnetic characteristics. When a lanthanide ion is exposed to a light photon, it absorbs some of the light and emits other wavelengths because its 4*f* shell is not completely filled. When a photoluminescent agent such as SrAl_2_O_4_ is co-doped with Eu^2+^ and Dy^3+^ at a certain ratio, the afterglow period is increased by a factor of ten [20]. Adding Dy^3+^ to the Eu^2+^-doped strontium aluminate increases its luminescence and extends the afterglow effect to more than 15 h. It has been claimed that rare-earth ions have been added to several types of glasses for possible optical electronic applications. There have been a number of studies [21] conducted to determine whether it is possible to generate bright glass by doping with rare-earth cations. The most studied persistently emitting luminous compounds are lanthanide-doped aluminates (LA; MAl_2_O_4_; M = Sr, Ca, Ba). Changing the amount of LA present in the supporting medium affects the intensity of the afterglow. The luminous features of strontium aluminates doped with rare earth are noteworthy because of their high quantum yield, chemical stability, safety, prolonged afterglow duration, and dazzling pure hues [22,23,24,25,26]. Due to their exceptional photostability, lanthanide aluminates (LAs) exhibit high reversibility. Inorganic phosphors have shown different emission colors such as SrAl_2_O_4_:Eu^2+^;Dy^3+^ as greenish emitter [27], CaMgSi_2_O_6_:Eu^2+^;Dy^3+^ as bluish emitter [28], and Y_2_O_2_S:Eu^3+^;Mg^2+^;Ti^4+^ as reddish emitter [29]. These photochromic inorganic phosphors are among the most impressive colorants due to their efficient photoluminescence, persistent emissions (>10 h), and resistivity to light, heat and chemicals [30,31]. The benefits of alkaline earth aluminates include their lack of toxicity and radioactivity, which makes them an attractive option. LAs have been recommended for several applications [32,33,34] due to their photochromic and persistent phosphorescence characteristics. It has been observed that a change in the concentration of LAs can alter the photochromic and persistent phosphorescence properties of a bulk polyester resin. Therefore, the incorporation of Eu^2+^ and Dy^3+^ doped strontium aluminum oxide into a glass material represents a significant step toward the creation of transparent smart glass with long-lasting phosphorescence, energy-saving disposition, tough surface, photochromism, and cheap price [35]. Glass is characterized by its qualities of transparency, weather and rust resistance, waterproofness, and dustproofness. Thus, glass has made it possible to create windows that let in a lot of natural light. However, glass materials are brittle, costly, heat transparent, and unsafe for earthquake-prone areas [36]. On the other hand, a polyester resin can be described as an unsaturated synthetic resin that can be prepared from the interaction of a polyhydric alcohol with a dibasic organic acid. Polyesters have been utilized in various fields such as sheet and bulk molded products, fiberglass, adhesives, and cured-on-site pipes. Polyesters are distinguished by their resistance to aging, chemicals, and water. They are thermally stable up to 80 °C and cheap [37,38,39]. To boost the mechanical characteristics of polymer resins, researchers have produced fiber-reinforced polymeric composites by encasing fibers as fillers within a polymer bulk. Fiber-reinforced polymeric composites have found widespread uses in several industries, including the construction of satellites, vehicles, and aeroplanes. Nanofiber-reinforced polymer nanocomposites [40,41] have attracted a lot of attention as viable materials for many fields. This could be attributed to the improved interfacial binding strength of the filler with the matrix, which is much improved due to the increased surface area of the integrated nanofibers. Electrospinning technology has provided a practical method for producing electrospun nanofibers from a variety of materials, including ceramic, polymer, and carbon. For nanofiber-reinforced polymer nanocomposites, electrospun glass nanofibers based on silicon dioxide have been presented to have excellent mechanical properties [42].

Persistent photoluminescent, energy-saving and photochromic windows and concretes have only been described in limited studies. Herein, we have been inspired to develop new transparent GLS@PET materials with high light transmittance, photochromism, hydrophobic activity, UV protection, and long-persistent photoluminescence (lighting in the dark) to minimize energy usage in buildings. Electrospinning was used to prepare glass nanofibers to be immobilized together with LANPs across a polyester matrix, creating photoluminescent GLS@PET hybrids. TEM was used to examine LANPs. EDXA, SEM, and XRF were used to examine the morphologies of the prepared electrospun glass nanofibers and GLS@PET hybrid substrates at various concentrations of LANPs. The optical characteristics of the GLS@PET samples were investigated by looking at their luminescence spectra. Under UV illumination, GLS@PET became green, according to the colorimetric screening conducted by CIE Lab parameters. The scratch resistance of the GLS@PET substrates implanted with LANPs was shown to increase in tandem with the phosphor content. Studying the static contact angle revealed improved hydrophobic characteristics. For various possible uses, including safety warning, anticounterfeiting, and soft illumination, the present luminescent GLS@PET hybrids are photostable, and can provide smart window and concrete with transparent photoluminescence capabilities.

## 2. Experimental

### 2.1. Materials

Chemicals for Modern Building International (Egypt) supplied the polyester resin (189-frp; 99.9%) and the hardener (Methyl ethyl ketone peroxide; MEKP). The unsaturated polyester resin is a yellow, translucent fluid with a hardening time of 15–25 min. The hardening agent was purchased from Sigma-Aldrich (Egypt). Sigma-Aldrich (Egypt) supplied polyvinyl pyrrolidone (PVP), tetraethylorthosilicate (TEOS), 3-aminopropyl triethoxysilane (APTES), and 3-glycidoxypropyltrimethoxysilane (GPTMS). Merck (Egypt) provided the basic ingredients used to synthesize LANPs, including Eu_2_O_3_, H_3_BO_3_, Dy_2_O_3_, Al_2_O_3_, and SrCO_3_.

### 2.2. Synthesis of LANPs

The LA phosphor was created by the high-temperature solid-state technique [43]. A combination of 0.2 mol of Al_2_O_3_, 0.002 mol of Eu_2_O_3_, 0.02 mol of H_3_BO_3_, 0.1 mol of SrCO_3_, and 0.001 mol of Dy_2_O_3_ was stirred for 3 h in a 300 mL of 100% ethanol. The combination was subjected to an ultrasonic treatment for 1 h at 35 kHz, drying for 2 h at 90 °C, and milling for 3 h. Sintering at 1300 °C for three hours was applied to the resulting residue to provide the LA microparticles. The LANPs were produced by subjecting the provided phosphor microparticles to the top-down technique [44]. Ten grams of LA microparticles were put into a 20-cm-diameter ball milling tube (stainless steel) mounted on a vibrating disc. A silicon carbide (SiC) milling ball was used to grind LA powder in the tube by the continuous collision between the pigment-charged tube and the vibrating disc for 23 h, introducing LANPs.

### 2.3. Electrospinning and Silanization

The electrospun glass nanofibers were made using a solution of TEOS (14%) and PVP (14%) in a solvent mixture of dimethyl sulfoxide/dimethylformamide (1/2) and then pyrolyzed at 800 °C, as described in a previous publication [45]. For 10 min, the given glass fibers were suspended in distilled water (5%), and they were sonicated for 10 min. The fibers were then immersed in a solution of silane (15%) in absolute ethanol, heated to 50 °C, and stirred at 125 rpm for an hour to complete the silanization process. Ethanol was used to both homogenize the mixture and rinse it after it had been allowed to sit for 10 min. To remove moisture from the supplied glass nanofibers, they placed a desiccator under a vacuum.

### 2.4. Preparation of GLS@PET

Polyester resin was mixed with the glass fibers (5% *w*/*w*), stirred for 15 min at room temperature and 125 rpm, and then homogenized for 5 min to ensure a uniform distribution of the electrospun glass nanofibers throughout the polyester bulk material. LANPs were charged into the above solution at a range of different concentrations (0.5%, 1%, 2%, 4%, 6%, 8%, 10%, and 12% *w*/*w*). Different symbols, including LA_0_, LA_1_, LA_2_, LA_3_, LA_4_, LA_5_, LA_6_, LA_7_, and LA_8_, were used to represent the resultant nanocomposites, respectively. The ingredients were stirred for 15 min at 150 rpm and sonicated for 5 min. MEKP was added at a weight percentage of 1.5%. The combination was exposed to stirring for 5 min, subjected to drop-casting onto an aluminum mould, and cured for 30 min at room temperature to yield a composite panel measuring 200 mm in length, 5 mm in thickness, and 200 mm in width.

### 2.5. Characterization and Analysis

#### 2.5.1. Topographical Measurements

The LANPs’ morphology was examined by a JEOL 1230 TEM (Tokyo, Japan). Ten minutes of ultrasonic (35 kHz) treatment of LANPs in CH_3_CN were followed by adding a drop onto a copper grid for TEM analysis. The GLS@PET morphologies were verified using a Quanta FEG-250 SEM (Czech). An EDX (TEAM) system connected to a SEM was used to analyze the luminescent GLS@PET substrates for their elemental composition. Additionally, Axios sequential XRF (Axios Instruments, Netherlands) was used to examine the chemical composition of the GLS@PET substrate.

#### 2.5.2. Hydrophobicity Measurements

OCA15EC (Dataphysics, Filderstadt, Germany) was applied to determine the contact angle of the GLS@PET sheets [46].

#### 2.5.3. Mechanical Testing

The LANPs-embedded GLS@PET nanocomposite substrates were tested for their resistance to scratching [47] using HB pencils. They were also examined for their hardness properties [48] using a Shore D hardness tester (Otto Wolpert-Werke, GMBH, Ludwigshafen, Germany). The sample dimensions were a 55 mm diameter and a 20 mm thickness.

#### 2.5.4. Ultraviolet Blocking

The ultraviolet protection factor (UPF) [49] was used to determine the UV-shielding efficiency of the luminescent GLS@PET hybrid samples. UPF was measured employing a UV-visible spectrophotometer according to the AATCC 183(2010) standardized technique.

#### 2.5.5. Photoluminescence Analysis

A JASCO FP-8300 (Japan) was employed to run the spectral analysis of photoluminescence. The photoluminescence lifetime of the ready-to-use GLS@PET substrates was measured using a phosphorescence accessory tool. In order to investigate the decay time, the GLS@PET substrate was subjected to 15 min of UV irradiation. The GLS@PET sample was then completely shielded from the UV light source to measure the decay time results.

#### 2.5.6. Reversibility Evaluation

For five minutes, the photoluminescent GLS@PET sample was exposed to UV light as per the established protocol [50]. After waiting 60 min in a dark wooden box, the GLS@PET sample was returned to its original state. Multiple rounds of UV irradiation and measurement of the resulting fluorescence spectra were performed.

#### 2.5.7. Colorimetric Properties

UltraScanPro (HunterLab, Reston, VA, USA) assessed the CIE Lab coordinates and color strength (*K*/*S*) of GLS@PET substrates before and after UV irradiation. Known in French as the “Commission Internationale de L’éclairage,” the CIE Lab is widely considered to be the preeminent authority on issues related to color vision, lighting, and illumination. The CIE Lab system [51] provides a numerical depiction of colors. Before and after being subjected to UV light, the Canon A710IS was utilized to take images of GLS@PET.

## 3. Results and Discussion

### 3.1. Preparation of GLS@PET Sheets

The solid-state high-temperature synthesis [43] was employed to prepare LA micropowder, which was subsequently treated with the top-down grinding technology [44] to yield LA nanoparticles. Figure 1 displays TEM and selected area electron diffraction (SAED) images. Neither defects nor dislocations were detected in the SAED image [52]. Using TEM, the diameter of LANPs was measured to be between 7 and 15 nm. The nano-scale materials have been essential to achieve the transparency of a matrix, which is critical for a variety of applications [53]. Thus, the LANPs were used to maintain the transparency of the GLS@PET matrix. Several concentrations of LANPs were used to create GLS@PET hybrids. The glass fibers were electrospun using formerly described methods [45]. Different concentrations of LANPs were blended with the given electrospun glass nanofibers and polyester resin. The present persistent phosphorescent, hydrophobic, photochromic, and UV protective GLS@PET sheets have shown to be practical toward industrial production for a variety of applications, such as directional marks, soft lights, smart window, smart concrete, and anticounterfeiting materials.

### 3.2. Morphology Features

Figure 2 and Figure 3 show the topographical characteristics of glass fibers and luminous GLS@PET substrates, respectively. Using EDXA, the elemental contents (wt%) of GLS@PET at three dissimilar surface locations are shown in Table 1. SEM analysis of the manufactured electrospun glass nanofibers revealed diameters between 90 and 140 nm. The surface topography of GLS@PET was maintained unchangeable when the LANPs ratio was raised. No phosphor particles were monitored on the GLS@PET surface, indicating that LANPs are entirely incorporated within the GLS@PET bulk.

The EDXA analysis verified the presence of the phosphor elemental components in the GLS@PET substrates. The EDXA analysis of the GLS@PET surface was performed at three distinct sites to evaluate the elemental compositions of the fabricated GLS@PET substrates. Similar elemental ratios were detected at the three examined locations, demonstrating that the phosphor particles spread uniformly over the sample matrix. According to Table 1, many elemental components were detected by EDXA analysis. Due to the incorporation of LANPs and glass nanofibers in the polyester bulk, many elements were detected in the GLS@PET substrates, including oxygen, carbon, silicon, strontium, aluminum, dysprosium, and europium. Both oxygen and carbon were assigned to the polyester bulk material’s. The presence of silicon was assigned to the glass nanofibers. Low levels of strontium, europium, dysprosium, and aluminum were found due to the trace quantities of LANPs employed in the production process of the GLS@PET substrates.

XRF was also applied to determine the chemical compositions of the luminous GLS@PET sheets, as shown in Table 2. The EDXA method provides a more accurate elemental analysis of a material. However, the XRF analysis can only identify elements at quantities higher than 10 ppm [54]. Hence, XRF provides a diagnostic method for the partial determination of a material elemental composition. Thus, only aluminium and strontium were detected by XRF in the GLS@PET substrates with lower LANPs ratios (LA_0_ and LA_1_), and the minute amounts of Eu and Dy present were not traceable. Strontium, dysprosium, europium, and aluminum were identified by XRF in GLS@PET with higher LANPs ratios (LA_2_-LA_8_). EDX and XRF studies indicated that the elemental ratios in LA and luminous GLS@PET substrates were quite similar.

### 3.3. Photoluminescence Spectra

A translucent backdrop of a material substrate is necessary for better visual perception of the color change to green [55]. A phosphor-infused GLS@PET hybrid was tested for photochromism and found to undergo fast, reversible color change. The GLS@PET-containing LANPs with ratios of 1% or less showed instant reversibility, indicating fluorescence emission. The sheets of GLS@PET with LANPs content over 1% kept glowing even after being exposed to darkness, a phenomenon known as delayed reversibility that indicates afterglow emission. Figure 4 displays the excitation spectra of GLS@PET, illustrating the impact of LANPs’ concentration on the resulting sheets. The emission peak intensity of LA_6_ increased as the duration of UV illumination increased from 100 to 400 s. However, no more increments were detected in the emission peak intensity of LA_6_ upon increasing the illumination time above 400 s. Figure 5 depicts the LA_6_ photoluminescence spectra against the illumination time. The emission peak was monitored at 518 nm after being excited at 368 nm. The strength of the emission intensity band grew with the lengthening of the illumination time. The glass nanofiber-reinforced polyester served as a transparent trapping bulk for the phosphor pigment nanoparticles. By either physical entrapment of LANPs within the GLS@PET medium or by coordination binding between Al^3+^ of LANPs and the oxygen atoms of GLS@PET [50,55], the addition of LANPs strengthened the binding between GLS@PET polymer chains.

It has been observed that the 4*f* ↔5*d* transition of Eu^2+^ produces a green light emission wavelength of 518 nm. The LANPs usually provide two emission bands of blue (shorter wavelength) and green (longer wavelength) color emissions that originate from two different strontium locations in the SrAl_2_O_4_ crystal. However, the ambient thermal quenching considerably reduces the intensity of the blue peak [56]. As a result, we could only see emissions in a greenish color. Therefore, we may deduce that only the emissions from Eu(II), and not Eu(III), had an effect on the photoluminescence spectra. The time-dependent exponential decay of the light from the GLS@PET substrate is second-order. The rate of deterioration was initially rapid but slowed considerably afterwards.

### 3.4. Photochromic Study

Phosphor nanoparticles were incorporated into the GLS@PET hybrid to create a transparent smart sheet, such as a smart window or concrete. As shown in Figure 6, photos of LA_6_ were taken in both the visible spectrum and UV illumination to analyze the photochromism of the GLS@PET sheets. Significant bright green emissions were observed in the UV spectrum, but no traces were discernible in the daylight spectrum. Anticounterfeiting applications, such as packaging, have made use of photochromism, which is caused by exposure to UV light [57,58,59]. Therefore, any product that employs a standard anti-counterfeiting strategy may benefit from the current approach. A rectangular gasket was made using the existing GLS@PET hybrid. The used gasket is invisible during the day but becomes fluorescent green when subjected to ultraviolet illumination, making it hard to duplicate.

The current approach is a reliable strategy since it has been used successfully to develop several anti-counterfeiting solutions for a more robust marketplace. The created GLS@PET substrates had their optical transparency examined to verify their promised clarity. The optical transmittance somewhat reduced as the quantity of LANPs increased in the GLS@PET bulk. LA_1_ had a transmission rate of 91%, whereas LA_8_ only managed 86%. Under UV light, the LA_1_ and LA_8_ samples, which appeared clear in daylight, took on a distinctly greenish color. For the reason that the present photoluminescent GLS@PET hybrids are transparent during the visible spectrum, it is easier to produce anti-counterfeiting patterns to prevent the forging of commercial items. It has been hypothesized that the 4f^6^5D^1^ ↔ 4f^7^ transition of Eu(II) is accountable for the emission of the LA phosphor [51]. Due to the absence of a discernible emission peak for Eu^3+^ or Dy^3+^, it was concluded that Eu^3+^ had been substituted with Eu^2+^. Dy^3+^ was also shown to be responsible for the formation of traps that allow for the release of light in the dark and consequently allow for Eu^2+^ to revert to its ground state. To achieve durability and photostability, reversibility is required for materials with photochromism and persistent luminescence. Several cycles of color change under UV and visible lights were used to confirm that LA_6_ has maintained its excellent reversibility, as shown in Figure 7.

### 3.5. Colorimetric Studies

Table 3 displays the photo-induced chromic features of the produced GLS@PET hybrid composites. From LA_0_ through LA_6_, the GLS@PET substrates were clearly transparent. However, hybrid composites with a higher concentration of phosphor nanoparticles (LA_7_ and LA_8_) had a slightly white color. Only if the LANPs are evenly dispersed throughout the GLS@PET bulk can the transparency of phosphor-containing GLS@PET substrates be guaranteed, as nanomaterials have shown exceptional transparency [53]. The colorless GLS@PET substrates (LA_1_ through LA_2_) with negligible amounts of phosphor nanoparticles emitted an intense green color only under the ultraviolet spectrum, indicating fluorescence emission. When exposed to ultraviolet light, GLS@PET hybrid composites (LA_3_ through LA_8_) with high concentrations of phosphor nanoparticles emitted a greenish color below ultraviolet light and a greenish-yellow glow in darkness, signifying an afterglow. As the ratio of LANPs was raised, the GLS@PET hybrid turned from transparent to slightly white. While the LANPs ratio was raised from LA_0_ to LA_6_, the colorimetric strength value changed by a small amount under visible light. When the LANPs ratio was raised from LA_7_ to LA_8_ to signify transparency, a substantial rise in *K*/*S* was seen to signify a whiter color. When illuminated with ultraviolet, increasing LANPs from LA_1_ to LA_8_ led to higher *K*/*S*. The *K*/*S* values were larger under ultraviolet illumination than the equivalent non-illuminated GLS@PET. This could be attributed to the greenish emission below ultraviolet illumination in contrast to the colorless look measured below visible daylight. LANPs-free GLS@PET (LA_0_) showed little variation in the CIE Lab below either visible or UV illumination. In contrast, phosphor-containing GLS@PET substrates revealed significant variations for the CIE Lab. Increases in the LANPs ratio resulted in a reduction in light transmission, which in turn led to a marginal fall in L^*^ under visible lighting conditions. L^*^ was demonstrated to drop considerably under an ultraviolet device when the LANPs ratio was increased, indicating a better, greener shade. Slight changes in–a^*^ and +b^*^ were detected under daylight as the LANPs ratio increased. Under UV, the magnitudes of –a^*^ and +b^*^ were demonstrated to grow when the LANPs ratio was elevated. The concentration of phosphor nanoparticles in the GLS@PET hybrid composites (LA1 and LA2) was monitored for fluorescence emission. The GLS@PET substrates that contained the highest ratios of phosphor nanoparticles (LA_3_ and LA_8_) had afterglow emission. The values of +b^*^ and –a^*^ decreased when the quantity of LANPs was increased from LA_1_ to LA_8_. Higher quantities of phosphor produced a white hue in both the LA_7_ and LA_8_ substrates. The LA_2_ sample, which appeared colorless in visible light, exhibited the greatest amounts of photo-induced greener fluorescence and photochromic activity. On the other hand, the LA_6_ sample retained its colorless look despite having the maximum phosphorescent greenish emission and the ability to glow in the dark.

### 3.6. Hydrophobic and UV-Blocking Properties

When LANPs were introduced to the GLS@PET hybride, the contacting angle rose from 136.5° in the control sample (LA_0_) to 137.6° (LA_1_). When the phosphor ratio was raised from LA_1_ to LA_6_, the LA nano-scaled particles roughened the surface, which increased the contact angle from 137.6° to 146.2°, respectively. The roughness and contact angle of LA_7_ and LA_8_ were marginally reduced when the quantity of LA nanoparticles was increased. Smart windows that can filter ultraviolet rays are a convenient way to protect humans from sunburn, erythema, and skin cancer [28,29]. Figure 8 displays the results of testing the ultraviolet protection properties of GLS@PET. The integrated LANPs in LA_1_ have a significant ultraviolet absorption capacity, providing strong UV protection. Therefore, LA_1_ performance in terms of protecting against UV rays was much better than that of LA_0_. Increasing the LANPs concentration also improved the UV blocking properties of the luminous colorless GLS@PET hybrids. The transparent GLS@PET hybrid that emits light can be used as smart windows that reduce energy costs. During the day, a lot of UV light enters via the photochromic window. Therefore, it generates a greenish hue that blocks the sun’s rays from entering the building. The photochromic GLS@PET hybrid reverts to its colorless state in low light, allowing more light to enter the building interior.

### 3.7. Mechanical Properties

The hardness performance of the prepared GLS@PET samples is a critical factor in identifying their durability and extent of deformation. Therefore, hardness is an important character and a valuable parameter to evaluate a composite’s performance [48,55]. The aim of this study is to develop a method for making transparent GLS@PET with a smooth exterior. Consequently, a series of scratch and hardness tests were performed to evaluate the mechanical features of GLS@PET. As a quick and easy method, the scratch resistance property was evaluated using a pencil [47]. Scratch pencils (6B to 9H) were used to create scratch patterns on the GLS@PET hybrid composites. The LANPs-free GLS@PET sample (LA_0_) was easily scratched using the HB pencil. For the samples from LA_1_ to LA_8_, the scratch resistance values were monitored at H, H, H, H, 2H, 2H, 3H, and 3H, respectively. Thus, increasing the LANPs ratio improved the scratching resistance of GLS@PET. Figure 9 shows the relationship between the LANPs ratio and the hardness properties of GLS@PET. The hardness of the prepared GLS@PET samples was found to decrease from 12.96 kg/mm^2^ to 10.21 kg/mm^2^ with an increasing LANPs filler ratio from 0.5% (LA_1_) to 4% (LA_4_), respectively. The hardness was then increased up to 12.39 kg/mm^2^ at LANPs ratio of 12% (LA_8_). Likewise, the impact strength (per unit volume) was observed to decrease from 13.26 MPa to 9.14 MPa when increasing the LANPs ratio from 0.5% (LA_1_) to 4% (LA_4_), respectively. The performance of impact strength was then increased to 15.13 MPa at a LANPs ratio of 12% (LA_8_). The improved hardness could be attributed to the incorporation of LANPs that serve as a very effective stress transmission agent inside the GLS@PET framework. Increasing the phosphor ratio increases the GLS@PET hardness by strengthening the intermolecular coordination linkages between polyester oxygen and Al^3+^ of LA. Al^3+^ may function as a catalytic agent to increase the polyester polymerization rate, improving the sample hardness. The LA phosphor creates a 3D polymer network with a higher molecular weight [48,50,55] due to Al^3+^ as a coordinating crosslinker between oxygen on polyester chains.

## 4. Conclusions

Using a GLS@PET hybrid host material and LANPs, this research set out to create a smart transparent windows and concrete that reacts to UV light. This GLS@PET substrate has photochromic and long-lasting luminescence properties, making it suitable for application in smart window and concrete technologies. The current technique can be utilized to make transparent GLS@PET hybrid nanocomposites that can change color and rely on photochromics to toggle their light transmission. With this simple technique, we were able to prove that photochromic GLS@PET substrates with desired characteristics, including transparency, photostability, UV protection, and hydrophobicity, are possible to produce. LANPs (afterglow and photochromic agents) were synthesized using the high-temperature solid-state method followed by the top-down milling technology. Transmission electron microscopy examinations revealed that the generated phosphor nanoparticles had sizes between 7 and 15 nm. Electrospinning was used to prepare glass nanofibers as a reinforcement agent. Both LANPs and glass nanofibers were uniformly embedded in transparent polyester resin, making them multifunctional photoluminescent materials. The GLS@PET substrates were analyzed for their morphological features using EDXA, XRF, and SEM. Luminescent GLS@PET showed photochromism by changing color from translucent to green below UV light, as measured by photoluminescence spectra and CIE Lab values. The contact angle of the GLS@PET hybrid nanocomposites increased from 136.5° to 146.2° when the LANPs ratio was raised. Scratch resistance and hardness were also shown to improve with an increase in the phosphor ratio. It has been reported that the use of a 1% phosphor ratio is optimal for fluorescence photochromism in GLS@PET, yielding a transparent material that emits a vivid green color when exposed to ultraviolet light. The colorless GLS@PET with a 6% phosphor ratio was found to have the optimum ratio for long-persistent phosphorescence. The GLS@PET hybrid composites showed excellent photostability.

## Figures and Tables

**Figure 1 polymers-15-00761-f001:**
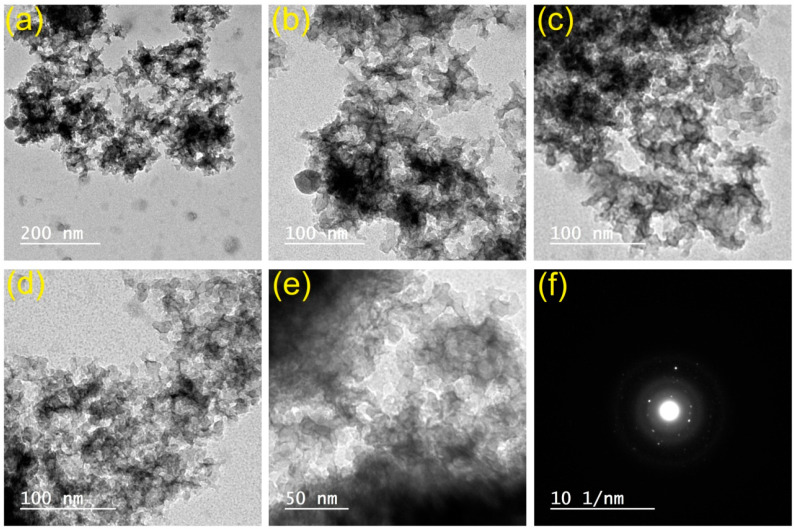
TEM (**a**–**e**) and SAED (**f**) images of LANPs.

**Figure 2 polymers-15-00761-f002:**
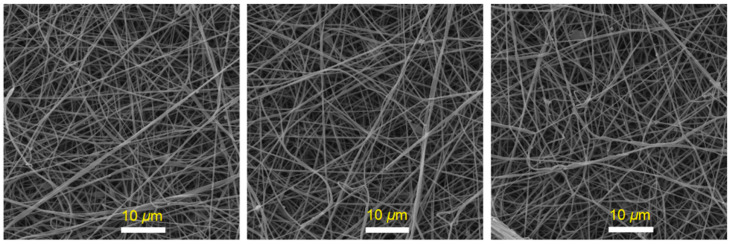
SEM images of electrospun glass nanofibers.

**Figure 3 polymers-15-00761-f003:**
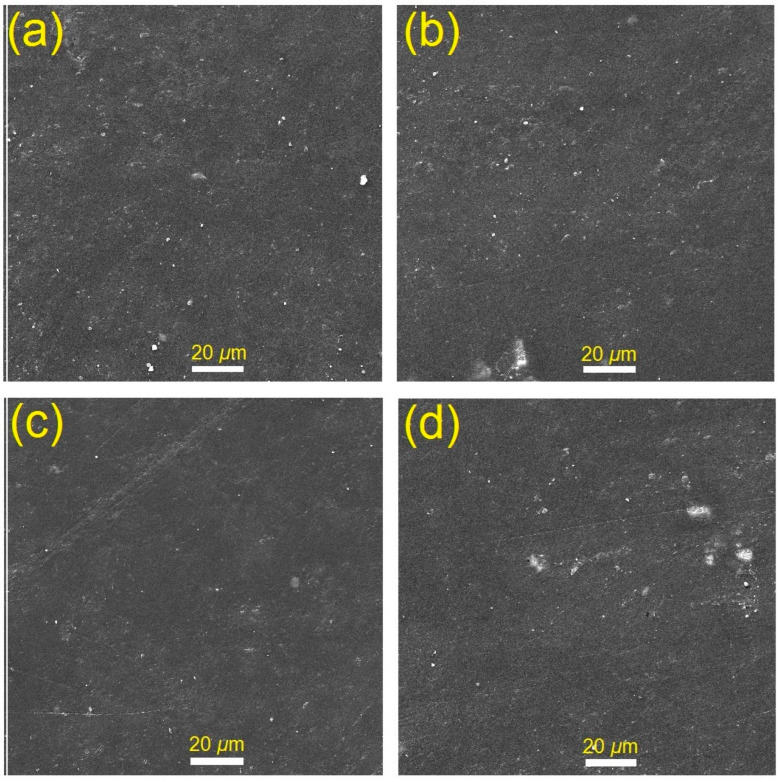
SEM images of LANPs-loaded GLS@PET; LA_6_ (**a**–**d**).

**Figure 4 polymers-15-00761-f004:**
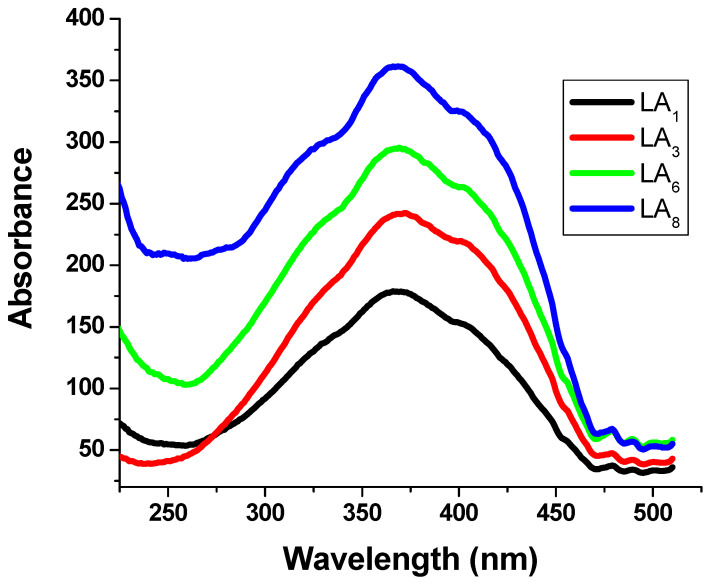
Excitation spectra of LANPs-loaded GLS@PET at 518 nm.

**Figure 5 polymers-15-00761-f005:**
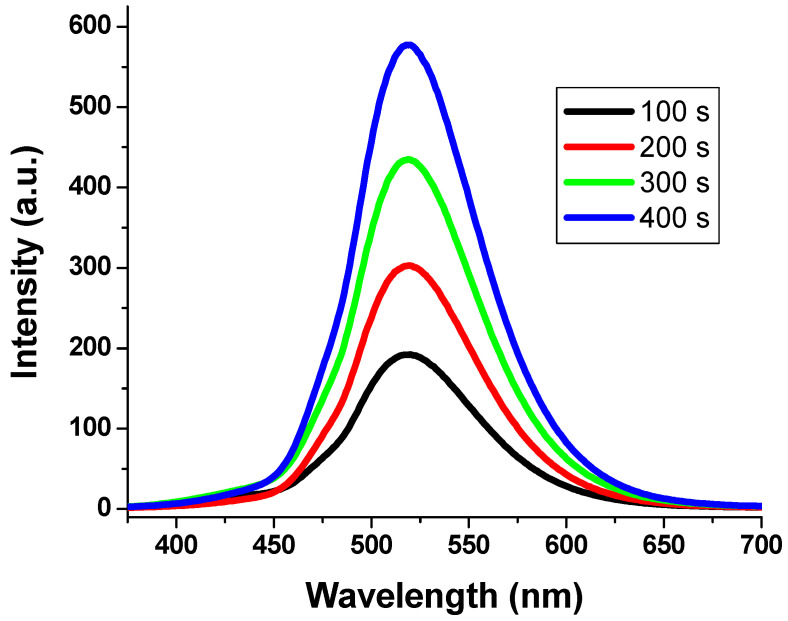
Emission spectra of LA_6_ with increasing the duration of ultraviolet illumination (100–400 s).

**Figure 6 polymers-15-00761-f006:**
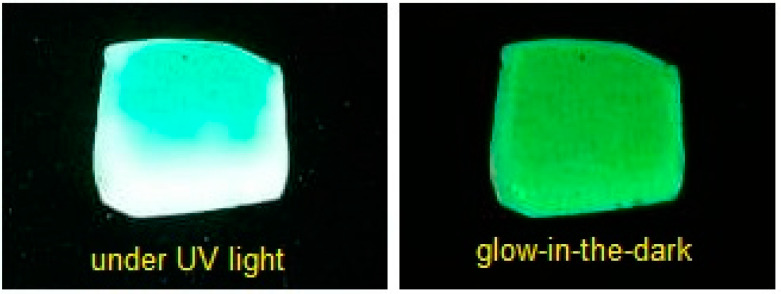
Photographs of LA_6_ bellow UV illumination, and under darkness.

**Figure 7 polymers-15-00761-f007:**
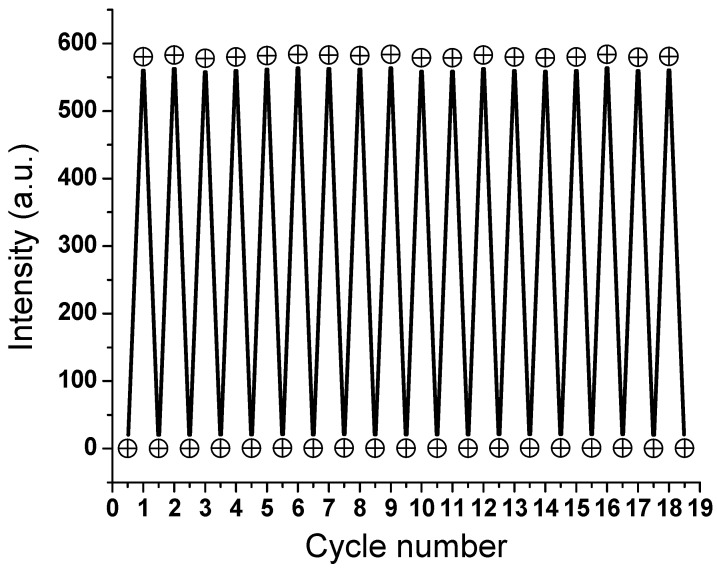
Reversibility of LA_6_ (518 nm) below UV and visible lights.

**Figure 8 polymers-15-00761-f008:**
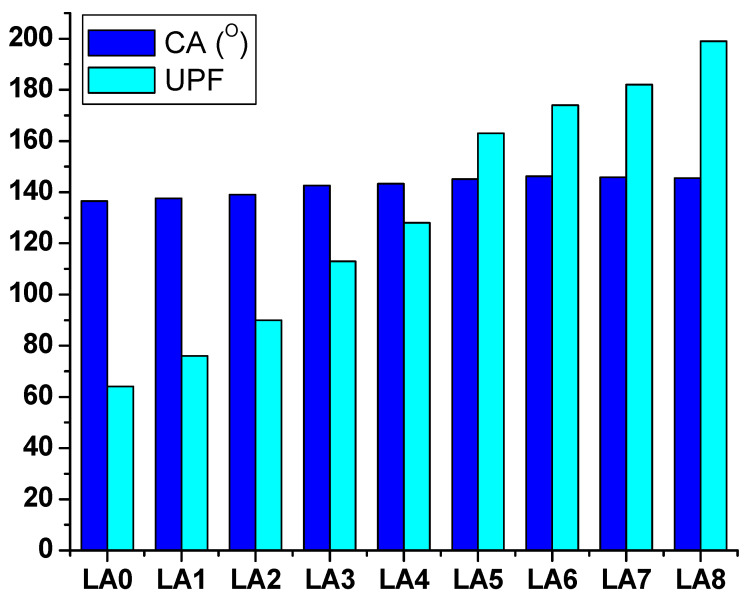
Contacting angle (CA) and ultraviolet protection factor (UPF) of GLS@PET.

**Figure 9 polymers-15-00761-f009:**
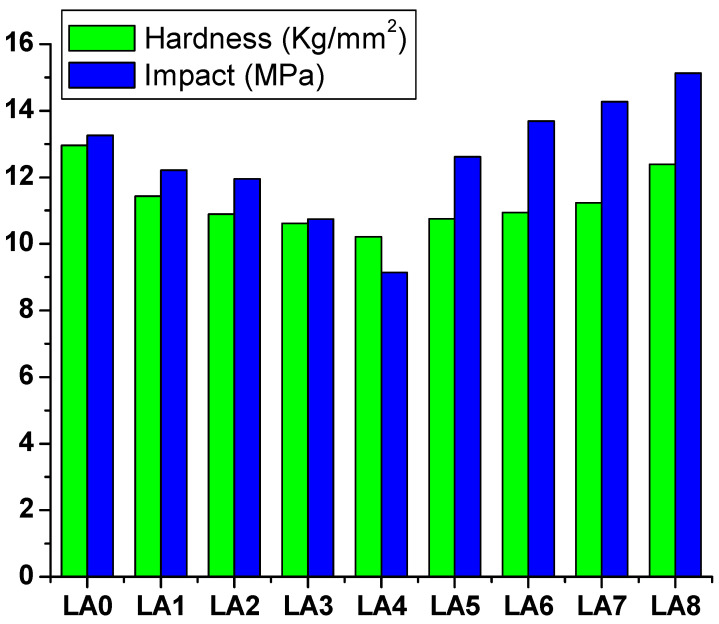
Hardness and impact strength of GLS@PET versus LANPs concentration.

**Table 1 polymers-15-00761-t001:** Elemental compositions (wt%) of LANPs-free and LANPs-loaded GLS@PET nanocomposites as detected by EDXA at three positions (S_1_, S_2_ and S_3_) on the GLS@PET surfaces.

GLS@PET		C	O	Si	Al	Sr	Eu	Dy
LA_0_	S_1_	57.09	37.25	5.73	0	0	0	0
	S_2_	57.06	37.25	5.74	0	0	0	0
	S_3_	57.17	37.01	5.91	0	0	0	0
LA_1_	S_1_	53.19	38.33	4.72	1.56	1.18	0.34	0.04
	S_2_	53.24	38.32	4.61	1.38	1.00	0.27	0.08
	S_3_	53.54	38.21	4.22	1.63	1.02	0.21	0.07
LA_3_	S_1_	52.37	39.18	3.65	3.94	2.31	0.37	0.15
	S_2_	52.86	39.00	3.25	3.84	2.23	0.49	0.22
	S_3_	52.13	39.31	3.64	3.80	2.43	0.40	0.21
LA_6_	S_1_	47.21	40.01	2.12	5.69	3.63	0.95	0.40
	S_2_	47.42	40.22	2.23	5.34	3.43	0.95	0.53
	S_3_	47.29	40.09	2.43	5.46	3.40	0.89	0.52
LA_8_	S_1_	44.22	41.45	1.33	6.58	4.56	1.06	0.81
	S_2_	43.90	41.36	1.53	6.80	4.33	1.19	0.72
	S_3_	44.46	41.62	1.37	6.55	4.19	1.23	0.74

**Table 2 polymers-15-00761-t002:** Elemental compositions (wt%) of LANPs-loaded GLS@PET as detected by XRF.

Element	Elemental Contents			
	LA_1_	LA_3_	LA_6_	LA_8_
Si	98.16	96.40	89.21	83.43
Al	1.17	2.15	6.23	9.27
Sr	0.67	1.21	3.13	5.25
Eu	0	0.41	1.32	2.50
Dy	0	0.23	0.75	1.62

**Table 3 polymers-15-00761-t003:** Colorimetric screening of LANPs-free and LANPs-loaded GLS@PET below visible spectrum (VS), and UV illumination; L* represents lightness between black(0) and white(100), a* represents color ratio between red(+a*) and green(−a*), and b* represents color ratio between yellow(+b*) and blue(−b*).

GLS@PET	Color Strength		L*		a*		b*	
	VS	UV	VS	UV	VS	UV	VS	UV
LA_0_	0.52	0.67	92.25	92.03	−1.41	−1.31	1.52	1.87
LA_1_	1.41	2.52	89.54	85.42	−1.27	−7.32	6.30	23.70
LA_2_	1.54	2.90	89.36	85.59	−1.35	−8.27	6.24	22.50
LA_3_	1.58	3.41	89.80	83.09	−1.20	−11.10	6.91	19.03
LA_4_	1.70	3.85	88.58	81.24	−1.82	−15.61	6.95	16.17
LA_5_	1.81	4.03	87.18	79.43	−1.87	−18.91	6.76	12.46
LA_6_	1.83	4.57	87.32	77.84	−1.69	−20.74	6.03	9.35
LA_7_	1.95	4.96	86.27	74.31	−0.86	−21.96	7.99	8.52
LA_8_	2.02	5.37	86.73	71.57	−0.58	−22.47	8.52	7.61

## Data Availability

Data Available from the corresponding author upon request.

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
