# Peer review of "Development of Photoluminescent and Photochromic Polyester Nanocomposite Reinforced with Electrospun Glass Nanofibers"

_polymers, 2023, doi:10.3390/polym15030761_

Round 1

Reviewer 1 Report

The present article has focused on developing photoluminescent and photochromic polyester nanocomposite, which showed promising results for practical use in smart windows and concrete, as well as anticounterfeiting patterns, so the article is relevant and interesting. The obtained nanocomposite has been studied by various methods, and the main conclusions of the article were confirmed experimentally. My comment is listed below. 

 The text style of the manuscript should be unified. Different font sizes were used, for example, on pages 2, 3, and 11, and it should be corrected.

Author Response

Reviewer #1:

Comments and Suggestions for Authors

The present article has focused on developing photoluminescent and photochromic polyester nanocomposite, which showed promising results for practical use in smart windows and concrete, as well as anticounterfeiting patterns, so the article is relevant and interesting. The obtained nanocomposite has been studied by various methods, and the main conclusions of the article were confirmed experimentally. My comment is listed below. 

The text style of the manuscript should be unified. Different font sizes were used, for example, on pages 2, 3, and 11, and it should be corrected.

Response: Thanks for the reviewer recommendations. The text style of the manuscript was unified.

Reviewer 2 Report

The manuscript titled “Development of photoluminescent and photochromic polyester nanocomposite reinforced with electrospun glass nanofibers” reports the preparation and development of a long-lasting photochromic and photoluminescent substrate with potential applications in the smart window, concrete, and anticounterfeiting patterns. The work appears to be interesting, but it suffers from several drawbacks and requires major revisions before publication, as follows:

1. Some typos and grammatical mistakes in the manuscript need to be refined thoroughly.

2.      In the abstract, the authors stated that “Transmission electron microscopic analysis (TEM) was utilized to examine the morphological features of lanthanide-doped aluminate nanoparticles (LANPs) and glass nanofibers to display diameters of 7-15 nm and 90-140 nm, respectively”. At the same time, there are no TEM results for glass nanofibers.

3.      In the Introduction, the authors should briefly describe the challenges in this field, discuss the previous reports on the topics of the manuscript, and finally declare the consequence of current results over previous reports (especially recently published work by authors, DOI: 10.1002/bio.4150 and  DOI:10.1002/bio.4333).

4.      It is suggested to write a paragraph about the optical features (photoswitching rate and time, photobleaching rate, photofatigue resistance, and …) of organic photochromic dyes (diarylethene, spiropyran, and …) and compare them with inorganic ones. The following references are suggested to be used, DOI: 10.1002/adfm.202007957 and DOI: 10.1016/j.jphotochemrev.2022.100487

5.      It is difficult to recognize the formation of LANPs and estimate their morphology and size in Figure 1. I would suggest that the authors provided TEM images with higher magnification and resolution. In addition, the indexed ring at diffraction planes in the SAED image of LANP should be described in the manuscript.

6.      It has been shown in “Figure 5” that by increasing the UV illumination from 100 to 400 s, the emission intensity at 518 nm follows an increasing trend. Where is the optimum UV illumination time to achieve higher emission?

7.      The rate and time of spontaneous dark photoluminescence bleaching of GLS@PET hybrids at low concentrations of LANPs should be measured (quantitively) and compared with those for GLS@PET substrates with the highest phosphor concentrations.

Author Response

Reviewer #2:

Comments and Suggestions for Authors

The manuscript titled “Development of photoluminescent and photochromic polyester nanocomposite reinforced with electrospun glass nanofibers” reports the preparation and development of a long-lasting photochromic and photoluminescent substrate with potential applications in the smart window, concrete, and anticounterfeiting patterns. The work appears to be interesting, but it suffers from several drawbacks and requires major revisions before publication, as follows:

  1. Some typos and grammatical mistakes in the manuscript need to be refined thoroughly.

Response: Manuscript language was revised by all authors and a native English speaker.

  1. In the abstract, the authors stated that “Transmission electron microscopic analysis (TEM) was utilized to examine the morphological features of lanthanide-doped aluminate nanoparticles (LANPs) and glass nanofibers to display diameters of 7-15 nm and 90-140 nm, respectively”. At the same time, there are no TEM results for glass nanofibers.

Response: The authors apologize for such unintended typo. The statement was revised to “Transmission electron microscopic analysis (TEM) and Scanning electron microscopy (SEM) were utilized to examine the morphological features of lanthanide-doped aluminate nanoparticles (LANPs) and glass nanofibers to display diameters of 7-15 nm and 90-140 nm, respectively.”.

  1. In the Introduction, the authors should briefly describe the challenges in this field, discuss the previous reports on the topics of the manuscript, and finally declare the consequence of current results over previous reports (especially recently published work by authors, DOI: 10.1002/bio.4150 and  DOI:10.1002/bio.4333).

Response: Thanks for the reviewer recommendations. The introduction part was revised accordingly.

  1. It is suggested to write a paragraph about the optical features (photoswitching rate and time, photobleaching rate, photofatigue resistance, and …) of organic photochromic dyes (diarylethene, spiropyran, and …) and compare them with inorganic ones. The following references are suggested to be used, DOI: 10.1002/adfm.202007957 and DOI: 10.1016/j.jphotochemrev.2022.100487

Response: The optical features of organic photochromic dyes were discussed and compared to inorganic ones. The majority of commercialized photochromic agents are organic dyestuff such as Spiropyrans and diarylethenes [15, 16]. However, there are certain downsides to organic photochromic colorants, including high costs and poor photostabilty. Because the photochromism of such organic dyestuff is based on the molecular switching of their structure, their inclusion in bulk materials limits their molecular switching ability due to steric hindrance effects [17]. This means that the photostability of photochromic organic dyes can be diminished with extended exposure to UV light. However, inorganic photochromic agents are resistant to steric effects as they do not depend on a change in their molecular structural system. Thus, inorganic photochromic agents provide better photochromism of higher photostability as compared to organic agents [18]. Eu2+ and other activation lanthanide ions are well-known photon emitters, particularly stimulated by UV light [19]. Inorganic compounds with photoluminescent lanthanide activators have been found to exhibit novel optical, electrical, and magnetic characteristics. When a lanthanide ion is exposed to a light photon, it absorbs some of the light and emits other wavelengths because its 4f shell is not completely filled.

  1. It is difficult to recognize the formation of LANPs and estimate their morphology and size in Figure 1. I would suggest that the authors provided TEM images with higher magnification and resolution. In addition, the indexed ring at diffraction planes in the SAED image of LANP should be described in the manuscript.

Response: Thanks for the reviewer recommendation. The authors apologize and confirm that the current TEM instrument (available at National Research Centre, Egypt) has limited resolutions due to the poor maintenance. Figure 1 displays TEM and selected area electron diffraction (SAED) images. Neither defects Neither defects nor dislocations were detected in the SAED image. The spacing of the adjacent lattice planes was ca. 4.45 Å. This was compatible with the inter-planar spacing of (0 1 1) plane of monoclinic strontium aluminum oxide [52].

  1. It has been shown in “Figure 5” that by increasing the UV illumination from 100 to 400 s, the emission intensity at 518 nm follows an increasing trend. Where is the optimum UV illumination time to achieve higher emission?

 Response: Tanks for the reviewer recommendation. The emission peak intensity of LA6 increased as the duration of UV-illumination increased from 100 to 400 s. However, no more increments were detected in the emission peak intensity of LA6 upon increasing the illumination time higher than 400 s.

  1. The rate and time of spontaneous dark photoluminescence bleaching of GLS@PET hybrids at low concentrations of LANPs should be measured (quantitively) and compared with those for GLS@PET substrates with the highest phosphor concentrations.

Response: The photoluminescence bleaching of GLS@PET was already explored at various concentrations of LANPs as reported in section “3.3. Photoluminescence spectra” The GLS@PET-containing LANPs with ratios of 1% or less showed instantly reversibility, indicating fluorescence emission. The sheets of GLS@PET with LANPs content over 1% kept glowing even after being exposed to darkness, a phenomenon known as delayed reversibility to indicate afterglow emission. It was also reported in section “3.5. Colorimetric studies” that the colorless GLS@PET substrates (LA1 through LA2) with negligible amounts of phosphor nanoparticles emitted an intense green color only under ultraviolet spectrum, indicating fluorescence emission. When exposed to ultraviolet light, GLS@PET hybrid composites (LA3 through LA8) with high concentrations of phosphor nanoparticles emitted a greenish color below ultraviolet light and a greenish-yellow glow in darkness, signifying afterglow.

Reviewer 3 Report

I add the reviewer comment in the attachment.

Author Response

Reviewer #3:

Comments and Suggestions for Authors

We received paper from Polymers MDPI with the title “Development of photoluminescent and photochromic polyester nanocomposite reinforced with electrospun glass nanofibers”. The paper proposed a new method how to reinforced nanocomposite that reinforced with glass nanofibers. Before it goes to be published, several items need to be clarified.

  1. In the abstract, authors stated that they can improved superhydrophobicity of the materials. What is the meaning of “superhydrophobicity”? and what the different between “hydrophobicity” ?

Response: Thanks for the reviewer feedback. The static water contact angle for a hydrophobic surface is higher than 90°, whereas the static water contact angle for a superhydrophobic surface is higher than 150°. Because the higher static water contact angle detected in the current study was 146.2°, the authors fixed the terminology to “superhydrophobic”.

  1. In the introduction, glass fiber can be form as continuous and non-continuous version. Glass fiber also can be dongle with resin epoxy to make GFRP. Used the following reference to enrich introduction: “Failure Prediction and Surface Characterization of GFRP Laminates: A Study of Stepwise Loading”.

Response: Thanks for the reviewer recommendation. The mentioned article was cited in the manuscript.

  1. Please revised the manuscript font size and make it follow the rule from the MDPI template.

Response: The manuscript font size and format were revised according to the rule from the MDPI template.

  1. Add the diameter size of the fibers

Response: Freshly prepared glass fibers were re-studied by SEM and included in the manuscript.

  1. Changed to bar graph to make it easier to interpretated.

Response: Data illustrated in Table 4 were revised to bar graph (Figure 8).

  1. What is the name of UTM in here? Add the specification.

Response: The authors re-studied the mechanical properties of the prepared samples. Thus, it was better to explore their hardness. The LANPs-embedded GLS@PET nanocomposite substrates were examined for their resistance to scratching [47] utilizing HB pencils. They were also examined for their hardness properties using Shore D hardness tester (Otto Wolpert-Werke, GMBH, Germany).

  1. Add the sample dimension for the tensile loading

Response: The sample dimensions were a 55 mm diameter and a 20 mm thickness.

  1. Add the load displacement or stress-strain of the mechanical loading in the manuscript.

Response: The authors re-studied the mechanical properties of the prepared samples according to their hardness.

  1. Why in the Fig. 8 the results tend to stable (flat)?

Response: The hardness performance of the prepared GLS@PET samples is a critical factor to identify their durability and extent of deformation. Therefore, hardness is an important character and a valuable parameter to evaluate a composite performance [48, 55]. The aim of this study is to develop a method for making a transparent GLS@PET with a smooth exterior. Consequently, a series of scratch and hardness tests were performed to evaluate the mechanical features of GLS@PET. As a quick and easy method, the scratch resistance property was evaluated using a pencil [47]. Scratch pencils (6B to 9H) were used to create scratch patterns on the GLS@PET hybrid composites. The LANPs-free GLS@PET sample (LA0) was easily scratched using the HB pencil. For the the samples from LA1 to LA8, the scratch resistance values were monitored at H, H, H, H, 2H, 2H, 3H, and 3H, respectively. Thus, increasing the LANPs ratio improved the scratching resistance of GLS@PET. Figure 9 shows the relationship between the LANPs ratio and the hardness properties of GLS@PET. The hardness of the prepared GLS@PET samples was found to decrease from 12.96 kg/mm2 to 10.21 kg/mm2 with increasing LANPs filler ratio from 0.5% (LA1) to 4% (LA4), respectively. The hardness was then increased up to 12.39 kg/mm2 at LANPs ratio of 12% (LA8). Likewise, the impact was observed to decrease from 13.26 MPa to 9.14 MPa when increasing the LANPs ratio from 0.5% (LA1) to 4% (LA4), respectively. The impact performance was then increased to 15.13 MPa at LANPs ratio of 12% (LA8). The improved hardness could be attributed to the incorporation of LANPs that serve as a very effective stress transmission agent inside the GLS@PET framework. Increasing the phosphor ratio increases the GLS@PET hardness by strengthening the intermolecular coordination linkages between polyester oxygen and Al3+ of LA. Al3+ may function as a catalytic agent to increase the polyester polymerization rate, improving the sample hardness. The LA phosphor creates a 3D polymer network with a higher molecular weight [50, 55] due to Al3+ as a coordinating crosslinker between oxygen on polyester chains.

Round 2

Reviewer 2 Report

The authors have adequately revised the manuscript and it can be accepted in the current version.